# Manufacturing of Aluminum Matrix Composites Reinforced with Carbon Fiber Fabrics by High Pressure Die Casting

**DOI:** 10.3390/ma15093400

**Published:** 2022-05-09

**Authors:** Javier Bedmar, Belén Torres, Joaquín Rams

**Affiliations:** Department of Applied Mathematics, Materials Science and Engineering and Electronics Technology, ESCET, Rey Juan Carlos University, Mostoles, 28933 Madrid, Spain; javier.bedmar@urjc.es (J.B.); belen.torres@urjc.es (B.T.)

**Keywords:** metal matrix composites, aluminum, carbon fibers, carbon wovens, high-pressure die casting, infiltration

## Abstract

Aluminum matrix composites reinforced with carbon fiber have been manufactured for the first time by infiltrating an A413 aluminum alloy in carbon fiber woven using high-pressure die casting (HPDC). Composites were manufactured with unidirectional carbon fibers and with 2 × 2 twill carbon wovens. The HPDC allowed full wetting of the carbon fibers and the infiltration of the aluminum alloy in the fibers meshes using aluminum at 680 °C. There was no discontinuity at the carbon fiber-matrix interface, and porosity was kept below 0.1%. There was no degradation of the carbon fibers by their reaction with molten aluminum, and a refinement of the microstructure in the vicinity of the carbon fibers was observed due to the heat dissipation effect of the carbon fiber during manufacturing. The mechanical properties of the composite materials showed a 10% increase in Young’s modulus, a 10% increase in yield strength, and a 25% increase in tensile strength, which are caused by the load transfer from the alloy to the carbon fibers. There was also a 70% increase in elongation for the unidirectionally reinforced samples because of the finer microstructure and the load transfer to the fibers, allowing the formation of larger voids in the matrix before breaking. The comparison with different mechanical models proves that there was an effective load transference from the matrix to the fibers.

## 1. Introduction

Metal matrix composites (MMCs) reinforced with carbon fibers (Cf) are interesting materials as they combine the different properties of the metal and the reinforcement used, providing better toughness and high yield and tensile resistances [1]. Aluminum matrix composites (AMCs), other works specify more advantages such as lightness and high corrosion resistance [2,3]. There are several processing techniques to fabricate this kind of composite. Spark Plasma Sintering (SPS) has been used to fabricate composites with good bonding between matrix and reinforcement [4] and with high mechanical properties, as shown by Tan et al. in their study [5]. However, most of the methods used for the manufacture of these types of composites are in a liquid phase, i.e., the molten aluminum is poured on the fibers [6], because it is a fast processing route and it allows the attainment of near-net-shape pieces [7,8]

However, Ramos-Masana and Colominas [9] claimed that the carbon fibers do not wet by the molten aluminum at the temperatures used for casting aluminum composites (700–800 °C) as the wetting angle is higher than 90° between Cf and aluminum because of the low reactivity of carbon fibers with the molten aluminum. Sarina et al. [10] confirmed that this effect remains even after long-time exposures of the Cf to the molten aluminum at higher temperatures. In the case of high modulus carbon fibers, the lack of wettability is enhanced because the basal planes of the graphite structure are placed parallel to the surface; this provides a low energy surface that further reduces the reactivity and hence the wettability of the carbon fibers [11].

The most common routes followed to get the infiltration of the fibers have been the increase of processing temperature, the coating of the fibers, the hot-pressing technique, and the use of high infiltration pressure.

Increasing the temperature of the aluminum alloy has been used by several authors to increase the wettability of different ceramic reinforcements: SiC particles [12], BC_4_ [13], and Cf [14]. However, in Cf, when the temperature exceeds 900 °C, aluminum reacts with the fibers and increases their wettability, but the reaction product formed is Al_4_C_3_, which is brittle and tends to hydrolyze in humid environments, expanding and degrading the composites.

To improve wettability, coatings have been deposited on the carbon fibers. The coatings inhibit the contact between the molten aluminum and the fibers thus that the wetting system is the molten aluminum coating, which is favored in most cases. Some authors have used the electroless technique to deposit different metals on long and short carbon fibers, such as Ni [15,16] and copper [17]. Refractory materials have been also used because of their inertness, and also pyrolytic carbon around the fibers has been suggested [18] but the results obtained have not fully solved the wettability problems in dense fiber zones. Finally, the incorporation of Mg to the alloy used also improves the wettability of the systems because it tended to situate at the Cf–Al interfaces [19].

Another way to avoid the wettability problem is using the hot-pressing technique on fibers coated by electroless. Nour-Eldin et al. obtained good bonding between matrix and reinforcement in short times of production with this method [20]. These composites showed a good distribution of the reinforcement with increased hardness and toughness [21] and increased wear resistance [22].

Another method is the continuous infiltration of wire composites to fabricate double composites. In this technique, wires of reinforcements are impregnated in the molten metal in a continuous process, and, after that, composites are obtained by a casting process. In this case, Kientzl and Dobránszky claimed that this kind of composite has improved mechanical properties [23]. In addition, Kientzl et al. claimed that the infiltration of the composite wire was successfully achieved [24].

In addition, from the manufacturing perspective, another way to solve the low wetting of the carbon fibers is to apply an external pressure to force the wetting of carbon fibers by the liquid aluminum. In the case of carbon fiber preforms, the pressure threshold that must be surpassed to infiltrate the fibers preforms has been established in the 35–106 MPa range [25], and pressures even higher are needed to get the full infiltration of the carbon fibers meshes of wovens. The lack of spontaneous wetting of the fibers also implies that there would be a tendency to form defects in the material, particularly in zones with a high density of fibers.

The pressure required for the infiltration can be applied by many different routes. The most used ones are squeeze casting, gas pressure-assisted infiltration, and centrifugal infiltration. During squeeze casting, pressure is mechanically applied to the alloy used after pouring it on the carbon fibers [26]. In some cases, this can be made on preforms made of short carbon fibers [27]. However, this method usually gives rise to the formation of pores and voids, and subsequent processing such as rolling reduced their influence on the mechanical properties of the composites [28]. The effect of the pressure used has also been analyzed in different systems. Sukumaran et al. [29] established that a pressure of 100 MPa was required for the microstructural refinement and to get very low porosity in Al 2124-10%SiCp composites. This indicates that once the infiltration threshold has been surpassed, it is not needed to apply much higher pressures to avoid the formation of defects. Finally, the centrifugal infiltration phenomenon has been used by authors like Nishida et al. [30] on fibrous preforms and metallic coated short Cf preforms, showing that it is a valid route, although it may cause gradients in the composition of the alloys, both in the location of the reinforcement and in the distribution of alloying elements in the matrix, what occurs in several kinds of AMCs [31].

On the other hand, aluminum matrix composites have been fabricated as coatings to improve the surface properties of metals like aluminum. In this case, thermal spray is a method that has achieved aluminum reinforced with SiC coatings with improved hardness, low porosity, and good bonding between matrix and reinforcement [32]. This wettability and low porosity can be enhanced by applicating sol-gel coatings on the SiC particles before the thermal spray [33].

On the other hand, one of the most used casting methods for aluminum pieces is High-Pressure Die Casting (HPDC) [34]. In this technique, molten aluminum is injected into a metallic die, and before the cooling and solidification of the aluminum alloy, high pressure is suddenly applied by a piston driven by compressed nitrogen. The alloy solidifies quickly due to the heat evacuated by the metal mold, and a fine microstructure is obtained [35]. In addition, the applied pressure reduces the size and number of the defects, which in most cases are trapped gas bubbles from the lubricant used for the demolding of the piece [36]. This process allows obtaining aluminum pieces for very different applications with a very competitive cost and with high production rates [37]. Therefore, HPDC seems to be optimal for the infiltration of carbon fibers with no need for preforms since the molten aluminum and the reinforcement adapt to the mold. However, it has not been used in composite manufacturing until now because of the difficulties of placing the fibers, the need for highly complex machinery, and the low reinforcement rates that there seems to be possibly obtained.

There are many differences between the previous methods used and the one proposed in this work. The most important is the combination of pressure and time used. Pressure is much higher in the HPDC than in the other methods, except for squeeze casting. However, in HPDC the die is cold; thus, the process is much faster than in squeeze casting, resulting in much different structures and properties.

In this work, aluminum matrix composites reinforced with long carbon fibers have been manufactured by infiltrating continuous carbon fibers in a metallic die. The design of the die prevented the movement of the fibers and allowed the infiltration of the carbon fiber meshes. The microstructure and the presence of defects, as well as the distribution of fibers in the composite, have been studied. Microhardness measurements and tensile tests have been carried out on the manufactured samples, and several micromechanical models have been used to show the load transference from the matrix to the fibers.

## 2. Materials and Methods

The aluminum alloy used, supplied by Aluminio La Estrella, was the AA413 alloy (EN AC-44100), which is an alloy specifically designed for the die casting process. Its composition was (wt.%): 12.8 Si, 0.65 Fe, 0.55 Mn, 0.20 Ti, 0.15 Cu, 0.15 Zn, 0.10 Mg, 0.10 Ni, 0.10 Pb, 0.05 Sn, and Al-balance. The mechanical properties provided by the manufacturer were 150–170 MPa tensile strength; 1–2% elongation at break; and 75 GPa of Young’s modulus.

The carbon fibers used were AS4 and were supplied by Hexcel. They had an average diameter of 7.1 µm and were supplied as 2 × 2 twill wovens. Before the composite manufacturing, the fibers were heat-treated in an oven for 24 h at 175 °C to remove their sizing. In the case of the unidirectional reinforced composites, fibers were obtained by removing the fibers of the warp of the woven. This allowed having the same type of composites and effective reinforcement in the main direction of the samples, while in the transverse direction, there were substantial differences. The properties of the carbon fibers provided by the supplier were 4480 MPa tensile strength; 1.8% elongation at break; 231 GPa longitudinal Young’s modulus; and 13 GPa of transversal elastic modulus.

The procedure used to fabricate the samples is shown in Figure 1 and is based on using a high-pressure horizontal opening die casting machine. Different HPDC conditions were tested, and several evolutions of the mold were used before having the definitive manufacturing process. The die used had a dog bone shape with a length of 55.0 mm, a width of 10.0 mm, and a thickness of 2.0 mm in the central zone, the total length of the specimen was 116 mm. The carbon fibers were cut with a length of 150 mm and a width of 20 mm, and they were placed in the die to be used as preforms for the pressure infiltration process. The die had a system of cavities and bumps that allowed clamping of the fiber before closing the die and which kept it fixed and stretched during casting. To ensure the filling of the die, the mold included different thoughts, and the inlet was dimensioned to avoid the formation of defects caused by the solidification shrinkage characteristic of the aluminum-silicon alloys. The die was also preheated to 280 °C.

The molten aluminum was injected at 680 °C into the die with a piston. After the filling of the mold with liquid aluminum, a 200 bar pressure was applied to the aluminum through the piston by the expansion of compressed nitrogen. The external pressure was maintained until infiltrated aluminum solidified in the die. Finally, the composite material was extracted using 2 pushing elements that were placed at the wider zones of the samples manufactured.

Two types of carbon fiber preforms were used: unidirectional carbon fibers (UD); and a twill fabric with fibers with 0° and 90° orientation. Previously, different volume fractions were used to fabricate the composites by adding layers of carbon fiber. There was a limited infiltration of aluminum between the Cf layers causing defects, and the number and size of pores also increased. Therefore, only 1 layer of reinforcement was used to fabricate the composites. Table 1 summarizes the type of samples manufactured and the final proportion of carbon fibers in the samples.

After manufacturing, the samples were tested by X-rays. The microstructure was evaluated with an optical microscope (OM) equipped with a Leica ProPlus software to measure the porosity of the specimens and a scanning electron microscope (SEM) with the help of image analysis software. For it, samples were cut, embedded in conductive resin, and mechanically polished (up to 3 µm).

The grain size was determined from the obtained micrographs by optical microscopy, following the Equation (1):(1)$$d=\frac{C}{n_{L}\cdot M}$$
where *d* is the average diameter; *C* is a constant whose value is 1.5; *n_L_* is the number of grains per unit of length, and *M* is the magnification used.

Vickers hardness testing was carried out using a Microhardness Tester (SHIMADZU HMV-2TE) with applied loads of 980.7 mN (HV_0.1_) and 9.807 N (HV_1_) for 15 s in different zones of the cross-sections of the samples as well as to identify the hardening effect of the carbon fibers used. The number of measures per zone was 15.

Tensile tests were performed in a ZWICK/ROEL TYPE 8594.60 testing machine equipped with a contact extensometer. Elastic limit, tensile strength, and Young’s modulus of each kind of sample were evaluated. At least 10 samples of each kind of material were evaluated. Finally, the fracture surface of each sample was analyzed by scanning electron microscopy (SEM) using a Hitachi S-3400N microscope.

## 3. Results

### 3.1. Samples Manufactured

The samples were manufactured following the procedure indicated in the experimental section. Figure 2a shows one of the manufactured samples with the dog bone shape and with the throughs for the excess aluminum and the aluminum inlet. The samples did not show external defects, and no carbon fibers were present on the surface. This indicates that the carbon fibers were embedded in the aluminum alloy and that the fabric was not pushed away by the aluminum injected.

To evaluate the internal characteristics, the samples were evaluated by X-ray imaging. The X-ray test is not sensitive to carbon fibers, but voids or pores in the aluminum can be easily detected. The images obtained (Figure 2b) showed that there were no relevant or visible defects in the interior. Therefore, the aluminum filled the die and infiltrated the carbon fiber meshes.

### 3.2. Microstructure of Composites

The distribution of the carbon fibers in the composite was evaluated on the cross-section of the central zone of the samples. Figure 3 shows the OM images of the cross-section of the composites. The different orientations of the carbon fibers in the woven-reinforced composite can be observed in Figure 3a,b. The circular sections correspond to fibers that were perpendicular to the observed surface, while elliptical ones correspond to fibers that were oblique to the surface. The microstructure shows that the fibers maintained the woven order during casting. The order was also kept in the unidirectional composite (Figure 3c,d) and all the fibers maintained their original orientation.

The density of fibers in the images is higher for the woven (Figure 3a,b) than for the unidirectional composites (Figure 3c,d), as corresponds with the higher reinforcement rate used in the woven (8% vs. 4%).

The porosity measured was below 0.5%, which is a particularly good value even for die casting of aluminum pieces. Some small pores (below 30 µm) were observed in the zones where fibers agglomerated, but they were not predominant in the samples. No significant differences in terms of porosity or other defects were observed associated with the type of fabric used.

The carbon fibers were distributed all over the sample, but they tended to be located in the central zone of the sample. This was caused by the method developed for holding the carbon fibers during the manufacturing process and is also related to the use of preforms of carbon fibers that were much thinner than the hollow in the die.

The fibers in the woven were more aggregated than in the unidirectional composite. In the woven, the carbon fibers were intertwined in two directions, forming a 2D structure that reduces the mobility of the individual fibers during the casting process. On the other hand, the unidirectional fibers were looser and were more easily moved by the flow of molten aluminum during the injection of the metal through the casting process.

SEM images in Figure 4 show that aluminum surrounded the carbon fibers and that there were no voids or lack of wetting of the carbon fibers, showing that the Cf were well integrated into the composites.

The microstructures of the samples obtained by optical microscopy are shown in Figure 5. In all cases, they were mainly constituted by eutectic morphologies, although some dendritic α-Al zones could also be observed in some zones. This is caused by the high Si content of the AA413 aluminum alloy used.

In the microstructure of the alloy shown in Figure 5a, three different zones can be distinguished. Zone 1 corresponds to the zones of the alloy that are in the vicinity of the die; Zone 2 corresponds to those parts of the alloy that are farther from the mold but that are not highly reinforced with carbon fibers; and Zone 3 to those zones in which carbon fibers were preferentially located.

In Zone 1, the grain size of the alloy was 3.4 ± 0.2 µm (Figure 5b). This small value is characteristic of the HPDC manufacturing processes as the cold metallic die causes the rapid solidification and cooling of the aluminum alloy.

In Zone 2 a dendritic structure with a grain size of 22.0 ± 0.1 µm was observed (Figure 5c). The bigger grain size indicates that the alloy cooled at a lower rate than at the surface of the samples. It also indicates that in these zones, there were no solids that could act as nucleation points for the formation of the solid.

In Zone 3, i.e., the zones closer to the carbon fibers, the grain size was 9.0 ± 0.2 µm (Figure 5d,e). This size is smaller than in Zone 2 because the carbon fiber allows a faster solidification of the alloy due to its high thermal conductivity; also, they act as nucleation zones, as has been claimed by different authors with different reinforcements [38].

In the case of the unreinforced alloy, the grain size in the vicinity of the metal die was similar to that of the composite materials (3.6 ± 0.3 µm, as mentioned before). At the center of the samples, the grain size was 24.3 ± 0.1 µm (Figure 5d), which is greater than that observed in the composites, even in zones far from the carbon fibers. This indicates that the carbon fibers affect the cooling of the whole material, not only of the zones closer to them.

Apart from the eutectic and the dendritic α-Al zones, other precipitates appeared in the microstructure as bright zones (arrowed in Figure 5f) with diameters of about 5 µm. These precipitates correspond to the α-AlFeMnSi phase and are the result of the presence of Fe and Mn in the alloy used [39].

### 3.3. Hardness of Composites

The hardness of the composites was evaluated in the cross-section of the samples with a 0.1 and 1 kg loads. The hardness was HV_1_ 112 ± 2 and HV_0.1_ 111 ± 4 for the alloy; HV_1_ 141 ± 2 and HV_0.1_ 135 ± 5 for the woven reinforced composite; and HV_1_ 117 ± 1 and HV_0.1_ 124 ± 4 for the composite with unidirectional fibers. The presence of the carbon fibers increased the microhardness by 22% for the 8 vol.% Cf reinforced woven and by 12% for 4 vol.% Cf reinforced with unidirectional fibers, while the HV_1_ hardness was 4% greater in the unidirectional composite and 20% in the fiber woven reinforced composite. These results show that the reinforcement has an impact on the hardness of the composite that is greater as the reinforcement increases and that wovens increase hardness in all load ranges, presumably because of its bidirectional effect.

The changes observed indicate that the carbon fibers had a hardening effect. Apart from the higher hardness of the reinforcement, the reduction in grain size associated with the presence of fibers could have also increased the hardness of the aluminum matrix. This effect was observed by Gajalakshmi et al. [40] in aluminum matrix composites reinforced with fibers coated with copper and nickel.

The hardness varied across the cross-section of the samples (Figure 6). In the unreinforced alloy, the hardness was ~5% higher at the edges than in the central part of the samples. In the composites, the hardness was similar in the center and at the borders, with a variation of ±1% on both sides of the center.

The zones with the carbon fibers showed the highest hardness values, i.e., the central zones of the composites, due to the higher hardness of the fibers and their reinforcing effect. In addition, the zones with smaller grain sizes showed high hardness values, i.e., the edges of the samples. Small grain size increases the strength and hardness since grain boundaries act as blocking dislocation movement, as claimed by Inoue et al. [41].

### 3.4. Tensile Properties of the Composites

The different samples were tensile tested. Table 2 resumes the results from the tests, and Figure 7 represents the values for a better comparison of the results.

The yield strength of the die-casting aluminum was 177 MPa, the tensile strength was 216 MPa, and the elongation at break was nearly 1%. The strength values were above those indicated by the supplier, and the elongation at break was comparable. The values provided by the supplier should be valid for casting very different shapes and dimensions; thus, they suppose a lower limit for the material properties. In addition, the slender shape of the samples manufactured and the small grain size observed contributed to increasing the yield and tensile strength of the samples. Obtaining values that were similar to or above the references validates the manufacturing method used, as has been indicated by authors like Mahaviradhan et al. [42].

The yield strength of the composites was 199 ± 3 MPa for the woven-reinforced composite and 188 ± 6 MPa for the unidirectional reinforced. Therefore, the increase in the yield was 12% for the woven and 7% for the unidirectionally reinforced composite compared to the alloy. The tensile strength of the composites was ~275 MPa, which was 25% higher than the unreinforced. Finally, there was also an increase in the values of Young’s modulus. The values measured for the composites were ~84 GPa for the composites, while for the unreinforced alloy was 77 GPa.

The increase in Young’s modulus indicates that the load was being transferred from the alloy to the fibers, which is also supported by the increase in strength. The increase in the yield suggests that the fibers, along with smaller grain size, would limit the sliding of atomic planes and pin the defects of the alloys. The higher yield of the woven composite is explained by its higher carbon fiber content.

Elongation at break also provided some differences between the alloy and the composites. The elongation of the alloy was 0.84%, which is common in HPDC processes where defects and heterogeneity of the microstructure limit the ductility of the samples. In the composites, the values increased to 1.02% and 1.68% for the woven and unidirectional reinforcements, respectively. The differences indicate that the fibers withheld part of the load during the tests, delaying the failure of the matrix.

Both composites show higher elongations at break than the alloy, but there was no contribution from part of the fibers in the woven. It seems that those that were perpendicular to the tensile test direction acted more as defects and strain accumulating zones than as reinforcement.

Finally, the standard deviation of the tensile tests for the composites was similar to or lower than for the unreinforced alloy. The deviation arises from premature test failure, generally associated with the presence of porosity and defects. Therefore, a smaller deviation indicates that the samples were reproducible, and the quality of the materials was high.

## 4. Discussion

### 4.1. Wettability and Pressure Infiltration

The infiltration of aluminum alloys in carbon fibers finds different challenges. The first one is the penetration of the molten aluminum within the fiber meshes because the lack of wettability of the carbon fibers under 900 °C avoids the spontaneous filling of the interfiber spaces.

Apart from increasing the temperature, other alternatives have been explored to infiltrate the aluminum: modifying the composition of aluminum alloys by adding elements such as Mg as suggested by Landry et al. [43]; coating the carbon fibers with metals such as Ni [14] or Cu [44], but this results in the modification of the composition of the alloy or using higher pressures. Constantin et al. [45] used a gas pressure-assisted method that allowed applying pressures from 1.2 to 5 MPa, which allowed filling carbon preforms made of short carbon fibers. Squeeze casting processes used pressures from 30 MPa to 50 MPa but to avoid the solidification of the aluminum alloy, the mold and fibers were at 750 °C, and the slow cooling of the composite gave rise to thick microstructures. In addition, the productivity of the technique was very limited [26].

In our case, the infiltration pressure was applied by an HPDC system, in which the pressure was applied to the aluminum alloy by a piston that is firstly mechanically moved and finally impulsed by a gas. There were many differences between the previous methods used and the one proposed in this work. In HPDC infiltration, the pressure used was 200 bar, which allowed the infiltration of aluminum at 680 °C in a few seconds. In addition, unlike the other systems, a cold metallic die was used, which fastly cooled the molten aluminum, resulting in improved properties of the composite matrix, Figure 7.

### 4.2. Microstructure of the Samples

The HPDC process developed was capable of integrating the carbon fibers in the aluminum matrix. The high magnification images of the carbon fibers in the aluminum alloy (Figure 4) showed that there were no gaps between the matrix and reinforcement. This evidences that the pressure used was much higher than the minimum one required to overcome the lack of wetting of the carbon fibers with aluminum at 680 °C even for the short duration of the process.

Cf maintained their circular shape after the casting process (Figure 4), indicating that the carbon fibers were neither consumed nor degraded by their reaction with the molten aluminum. In addition, SEM images did not detect the presence of Al_4_C_3_. Lee et al. [46] observed the formation of Al_4_C_3_ and the degradation of the Cf in low-pressure infiltration at high temperature, and Yang and Scott [47] and de Sanctis et al. [48] observed the same reaction in squeeze casting. Aluminum carbide is brittle and reduces the mechanical properties of the composites, as Zhang et al. [49] reported. In addition, it is highly hygroscopic and, in the presence of humidity, expands and breaks, causing a strong degradation of the composite.

We have not observed either Al_4_C_3_ or the degradation of the carbon fibers, and this explains the excellent mechanical properties obtained for the composites. In HPDC the process temperature was kept low, slightly above the aluminum melting temperature, and the interaction time between the molten aluminum and the carbon fibers was very short.

The low interaction time was the second characteristic of the processing route used. The die was filled with aluminum in less than 0.5 s, and then high pressure was applied and maintained. The solidification of the aluminum took only a few seconds more. Simulations carried out in different conditions indicate that, even without carbon fibers, solidification times would be less than 4 s for the entire sample. These times were further reduced by the presence of carbon fibers, which also remove heat from the molten aluminum alloy. The short interaction time, together with the low temperature used in this process, prevented the formation of detrimental Al_4_C_3_. Wielage and Dorner [50] stated that composites with Al_4_C_3_ have higher corrosion than those without this phase.

In the microstructure, some phases constituted by Al, Fe, Mn, and Si were observed (arrowed zone in Figure 4b). The alloy used contained 0.65 wt.% Fe that was used to reduce the adherence of the samples to the mold, but it could promote the formation of acicular β-AlFeSi precipitates. These precipitates were not observed in the microstructure of the alloy because of the presence in the alloy of 0.55 wt.% Mn, which transformed the β-AlFeSi precipitates into the compact α-AlFeMnSi ones which were less detrimental to the properties of the material, as was explained by Lu et al. [51].

The composition of the alloy included Mg. The precipitates that Mg usually forms in 4xx alloys were Mg_2_Si and helped to increase the strength of the alloys. However, the alloy contained 0.1 wt.%, which makes its proportion small. In any case, their size is usually in the nanometer range; thus, special techniques would be necessary to determine their presence.

One important feature that requires analysis is the heterogeneous distribution of the carbon fibers in the composite. Fibers were preferentially located at the central zone of the transversal section of the samples because the cavity of the die was designed to be filled with aluminum on the fixed and mobile platen to avoid the presence of carbon fibers at the surface of the samples.

This distribution would have important effects on the corrosion and mechanical properties of the materials. The fibers were not present at the surface; thus, corrosion galvanic couples between carbon and the aluminum matrix were not formed. This solves the corrosion risks related to these composites, which are the cause of avoiding the combination of carbon fiber composites with aluminum bolts or pieces in the aeronautical and automotive industry. This is one of the causes of giving preference to the location of fibers inside the composite rather than forcing a homogenous distribution in the transversal section of the samples.

On the other hand, the preferential location of carbon fibers at the central zone of the cross-section can be a challenge because the properties of the composite are not homogenous across its section. These differences may cause the apparition of some added stress within the samples. In addition, the behavior of the composite would be affected by the uneven distribution of the carbon fibers.

To assess how critical this aspect could be, it is worth comparing it to carbon fiber-reinforced polymers (CFRPs). CFRPs are manufactured by stacking plies that are typically less than 0.2 mm thick in different orientations. The tensile strength of a layer in the direction of its fibers is very high. The resistance of a layer with fibers perpendicular to the applied load would be only that of the resin, which is very low [52]. However, the combination of layers that have such different properties has been accepted and used in CFRP laminates. In our system, the resistance parallel to the carbon fibers is the combination of the resistance of the carbon fibers added to that of the aluminum alloy, which has values of strength above those of the epoxy matrices used in many CFRPs. In the transversal direction, the resistance would be that of the aluminum alloy, which is still much higher than that of many polymers used in CFRPs. Therefore, the properties of the aluminum composites manufactured could be easily modeled, and the heterogeneous distribution of the carbon fibers should not result in a detrimental behavior of the composites for structural applications.

In addition, the aluminum matrix composites reinforced with unidirectional fibers should show different properties between the parallel and the transversal directions of the material, i.e., mechanical anisotropy, as the orientation of the carbon fibers determines the behavior of the composite. The anisotropy should be lower when using wovens as the reinforcement improves the properties in two perpendicular directions equivalently.

### 4.3. Mechanical Properties

#### 4.3.1. Modeling the Mechanical Properties

The composites developed have a higher Young’s modulus, yield, and maximum resistance than the alloy, indicating that load was transferred between the two phases of the composite, i.e., the aluminum matrix and the carbon fibers.

The Young’s modulus is similar in all aluminum alloys as it is not affected by the presence of different alloying elements or different tempering treatments. Therefore, the reinforcing effect of carbon fibers caused the observed change. We have used the rule of mixtures (ROM) and the Halpin–Tsai (HT) models to evaluate the effective reinforcement rate of the composites.

The ROM model considers that the two phases present in the composite, i.e., carbon fibers and aluminum matrix, behave with a perfect transference of load from one another. In this model, the final Young’s modulus value only depends on Young’s modulus of each phase and the corresponding volume fractions. The general equation for our system could be the one shown in Equation (2).
(2)$$E_{c}=E_{f}v_{f}+E_{m}v_{m}\;\;\;\mathrm{with}\;\;\;v_{f}+v_{m}=1$$
where *E_c_*, *E_f_*, and *E_m_* refer to the Young’s modulus of the composite, the fiber, and the matrix, respectively; *v_f_* and *v_m_* refer to the volume fraction of the fiber and the matrix, respectively.

This equation can be directly applied to the unidirectional reinforced composite as Young’s modulus of aluminum is isotropic, and carbon fibers are aligned in the direction of the tensile test.

In the case of the woven composite, the fibers that are in the longitudinal direction of the tensile tests (0°) have a different contribution from the ones placed in the transversal one (90°). In this case, although a laminated model could provide more precise results, the ROM also allows us to consider that we have three different phases as Equation (3) shows.
(3)$$E_{c}=E_{fL}v_{fL}+E_{fT}v_{fT}+E_{m}v_{m}\;\;\;\mathrm{with}\;\;\;v_{fL}+v_{fT}+v_{m}=1$$
where *E_f_*_L_ and *E_fT_* refer to the Young’s Modulus of the 0° fibers and 90° fibers, respectively; while *v_fL_* and *v_fT_* refer to the volume fraction of the fiber along the 0° and 90°, respectively.

The values measured for Young’s modulus were 85 GPa and 84 GPa for the unidirectional and bidirectionally reinforced composites, respectively. The Young’s modulus of the carbon fibers was 231 GPa in the fiber direction and 13 GPa in the transversal one, and its maximum resistance was 4480 MPa, as shown by Herráez et al. [53]. The ROM according to Equation (1) indicates that for 4% vol. of Cf Young’s modulus should be 83.2 GPa and in the case of the bidirectional reinforcement, the expected Young’s modulus was 89.3 GPa (Table 3). These results approximate to values measured; thus, it indicates that there was an adequate transference of load from the matrix to the fibers.

Islam and Begum [54] explained in their work the Halpin–Tsai model, which incorporates the effect of the interactions between fibers and matrix, and the form factor, i.e., the length (L) to diameter (D) rate, of the reinforcing fibers. Therefore, we assume that we have an ideal solid with no pores and that there is no interphase between matrix and reinforcement. Equation (4) can be directly applied to unidirectional composites and in it.
(4)$$E_{c}=E_{m}\left(\frac{1+\xi \eta v_{f}}{1-\eta v_{f}}\right)\;\;\;\mathrm{with}\;\;\;\eta =\frac{E_{f}/E_{m}-1}{E_{f}/E_{m}+\xi }\;\;\;\mathrm{and}\;\;\;\xi =\frac{2L}{D}$$

In our system, the form factor of the fibers was ~30,000; thus, the model approximates better than 0.5% to the ROM model, and the expected values nearly reproduce the ones from ROM, indicating an expected value for the unidirectional composite of 83.5 GPa.

In the case of the transversal fibers, two different approaches may be applied: (i) the transversal carbon fibers act as defects in the matrix; thus, it is like considering the presence of voids with a volume proportion corresponding to that of the transversal fibers; (ii) the transversal fibers act as very short fibers with a length similar to its diameter. The model is not valid for this second type of approximation, as the shape of the fibers is not adequate; thus, the first considered case has been included in Table 3. The value proposed by the model is 83.5 GPa for the unidirectional composite and 80.2 GPa for the bidirectionally reinforced composites. Both values are close to the experimental results, but we have not observed the reduction in Young’s modulus expected for the bidirectional reinforcement.

The models used fit the experimental results of the Young’s Modulus thus it can be determined that an effective reinforcement was obtained by the incorporation of unidirectional carbon fibers in the composite and that the presence of transversal fibers did not degrade Young’s modulus of the composite.

The combined results of a 7–12% increase in yield strength, 25% in tensile strength, and 10% in Young’s modulus indicate that the carbon fibers had an effect on the mechanical properties of the composite and that the load applied to the samples was partially transferred to the carbon fibers. Otherwise, the carbon fibers would have acted as defects in the composites and would have reduced the performance of the composites in the tests.

One relevant consideration is that, although the woven doubles the reinforcement rate of the alloy, Young’s modulus and the tensile strength are the same. This can be explained by the relative direction of carbon fibers with the direction of the stress applied to the samples during the tests. Only the carbon fibers in the test direction were capable of withholding load and, therefore, increasing Young’s modulus of the composite. The fibers that were perpendicular to the test direction contributed with their transversal Young’s modulus, which is substantially lower than the longitudinal one. Therefore, they could act as defects and lower the properties of the composite, but this effect has not been observed. In particular, the models used to calculate Young’s modulus in composites provide effective reinforcement rates of ~4.6%, which are very similar to the volume fraction of reinforcement aligned in the test direction.

In addition, considering the tensile resistance of the alloy and the carbon fibers, the maximum expected resistance for the composite would be 303 MPa, while the values obtained were ~276 MPa. Considering the most basic models for fiber-reinforced composites, it can be indicated that the effective reinforcement rate was in the range of 1.6–2.6%. This value is a minimum reference as the ripples of the fibers in the woven reduced the effectivity of the load transfer. Again, this indicates that most of the fibers in the load direction were effectively contributing to the stiffness of the composites.

The transference of load between matrix and reinforcement modified the hardness of the samples. Hardness has usually been related to the elastic yield of the alloys, and a proportional behavior has been observed. The woven reinforced composite showed a hardness increase of 22%, while the increase was 12% for the unidirectional composite. These values are similar to the observed increases in yield strength, which was 12% for the aluminum reinforced with woven and 6% for the unidirectionally reinforced one.

The hardness increase was not greater because of the orientation of the fibers. In the woven, half of the fibers were perpendicular to the cross-section where hardness was measured, but the other half was placed in the surface plane. Therefore, the effect of the fibers perpendicular to the indent was limited. In addition to the mechanical contribution of the fibers, the grain size distribution also had a strong effect on hardness. Therefore, the carbon fibers had a twofold contribution. On the one hand, Sree Manu et al. [55] claimed that they increase hardness as a result of their mechanical properties, particularly their high modulus and strength, and they act as a barrier to matrix dislocation. On the other hand, carbon fibers also form finer microstructures, like in other composites made by different techniques, as Aynalem [56] indicates, because their high thermal conductivity leads to the fast cooling and solidification of the alloy, while they act as pinning zones for the grains nucleation, resulting in a synergic increase of hardness in the alloy.

#### 4.3.2. Fractographic Analysis

Figure 8 shows the fracture surface of the different samples as observed by SEM. In all cases, the morphology corresponds to samples without casting defects. In the case of the alloy, cracks started by the clustering of microvoids (Figure 8a–c with orange arrows in the dimples). The fracture surface showed the characteristic dimples of the ductile fracture of many metallic materials, but due to the fine microstructure, their size was very small.

In the case of the samples reinforced with carbon fiber woven (Figure 8d–f), a more complex fracture mechanism was present. Cracks started at zones of the alloy that were not highly reinforced. Then cracks coalesced at the fibers that were approximately perpendicular to the tensile test direction. Moreover, the final fracture took place at zones with high content of carbon fibers. Due to this, the end of long fibers was visible protruding from the surface of the fractured samples (Figure 8e). In all cases, fibers were free of aluminum, and their shape was not modified by the HPDC process (Figure 8e).

The unidirectional reinforced composite fracture image (Figure 8g) shows that the cracks did not start with any defect in the structure. The coalescence of microvoids in the central zone of the samples was the cause of the fracture of the samples. Observing them with a higher magnification (Figure 8h,i) allowed determining the presence of many fibers sticking out of the surface of the fracture surface, as well as many holes left by the fibers that remained on the other side of the sample. This can be observed with more detail in Figure 8g in which some fibers protrude on the surface while some perfectly circular holes can be observed. In addition, in Figure 8i, it can be appreciated that the voids formed in the aluminum matrix were bigger than those observed in the unreinforced alloy.

Therefore, the carbon fibers played an important role in the fracture of the composites. On the one hand, they helped to increase the elongation at the break of the samples, particularly for the unidirectionally reinforced composite. In this case, the direction of the fibers was along the applied strain, which favored the higher elongation of the samples. According to the models shown, the 4 vol.% of Cf retained 30% of the load applied, thus the rest of the alloy evolved with less strain. The higher elongation observed is related to forming of larger voids in the matrix before breaking, favored by the fine microstructure.

In the case of the woven-reinforced alloys, the fibers that were in the 0° direction had the same effect as in the UD composite, supporting the load and helping for the deformation of the samples, but the ones oriented at 90° behaved differently. As they could not slide or retain load, they acted as barriers to deformation of the composite and as defects that helped fracture the samples. This causes the elongation at break in this configuration to be lower than in the UD composite. Despite this analysis, if the load had been in another direction, the woven-reinforced material would have had similar behavior, while the UD composite would have behaved similarly to that of the matrix or even worse.

## 5. Conclusions

The main conclusions derived from this work are the following:Aluminum matrix composites consisting of AA413 aluminum alloy reinforced with long carbon fibers in the shape of unidirectional and woven fabrics were manufactured by HPDC by incorporating the fabrics in the metallic die;Fibers were not degraded by the interaction with aluminum and most of the fibers remained in the zones where they were positioned before the entrance of the molten aluminum;The pressure applied by the HPDC process allowed the filling of the zones within the woven meshes and fibers, and the composites were practically free from defects. The carbon fibers limited the grain growth in the composites; thus, finer microstructures were obtained. As a result of this, hardness increased in the composites and more in the woven-reinforced composite because of its higher volumetric reinforcement fraction;The carbon fiber reinforcement used provided a 10% increase in Young’s modulus, a 6% with UD Cf and 12% with woven in yield strength, 28% increase in tensile strength, and the increase in elongation at break was 86% for the UD and 21% for the woven. The properties observed have been explained by the rule of mixtures models or by the Halpin–Tsai one, with a general deviation of less than 7%.Fractographic tests showed that the presence of fibers acted differently depending on the orientation. Fibers along the tensile test direction (0°) favored the ductile behavior of the alloy, and after the break, fibers arose from the fracture surface. In the woven-reinforced composite, the 90° fibers did not help to increase the strength of the composite and limited the deformation of the sample, therefore acting as defects.The material with the highest yield strength and tensile strength was the aluminum reinforced with woven with 8 vol.% Cf, while the material with the highest Young’s Modulus and elongation at break was the aluminum reinforced with 4 vol.% of unidirectional carbon fibers.

## 6. Patents

Marino Martín, P.L.; Arias Martín, R.; Carrero Hinojal, A.; Torres Barreiro, B.; Rams Ramos, J.; Ureña Fernández, A.; Sánchez Martínez, M.; López Galisteo, A.J.; Rodrigo Herrero, P.; Bedmar Sanz, J.; Mercado Sapia, C. Fabricación de materiales compuestos reforzados con fibra de carbono mediante inyección de una aleación de aluminio de alta presión. Spain Patent ES2802282, 18 May 2021.

## Figures and Tables

**Figure 1 materials-15-03400-f001:**
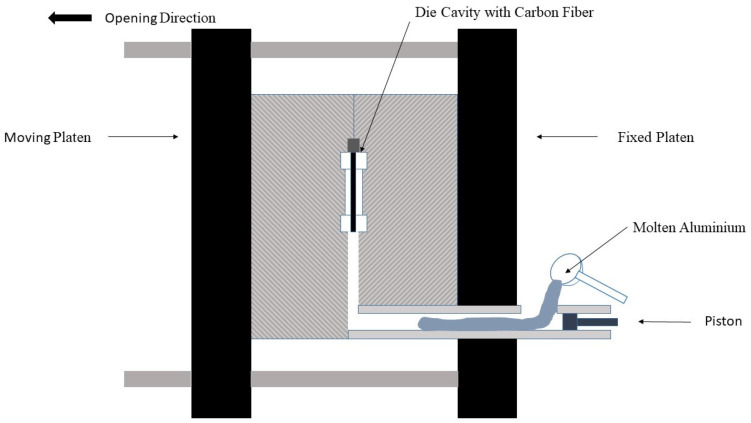
Scheme of the high-pressure die casting process.

**Figure 2 materials-15-03400-f002:**
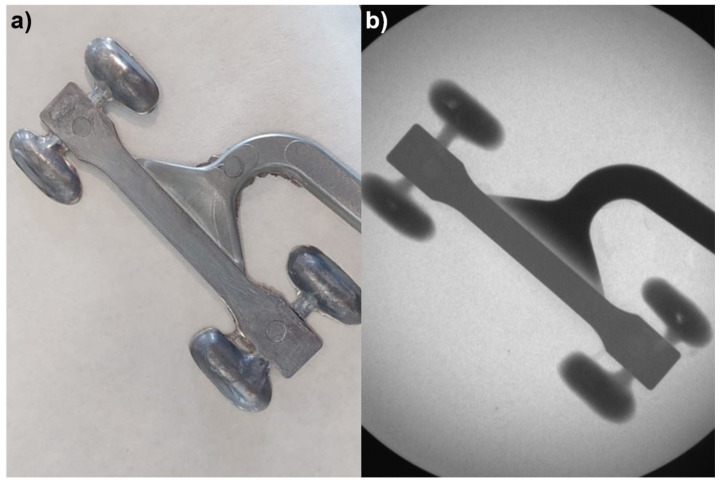
(**a**) As-manufactured composite sample, and (**b**) X-ray image of the sample.

**Figure 3 materials-15-03400-f003:**
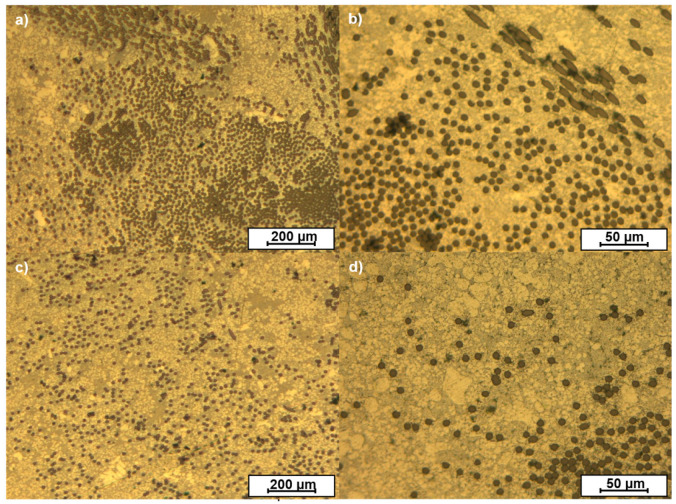
Optical micrographs of the AMCs: (**a**,**b**) samples manufactured with woven, and (**c**,**d**) samples manufactured with unidirectional fiber.

**Figure 4 materials-15-03400-f004:**
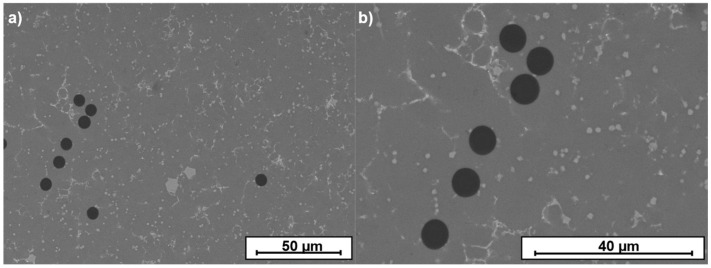
SEM images of the unidirectional carbon fibers in the Al matrix: (**a**) microstructure of aluminum reinforced with unidirectional carbon fiber; and (**b**) magnification of (**a**).

**Figure 5 materials-15-03400-f005:**
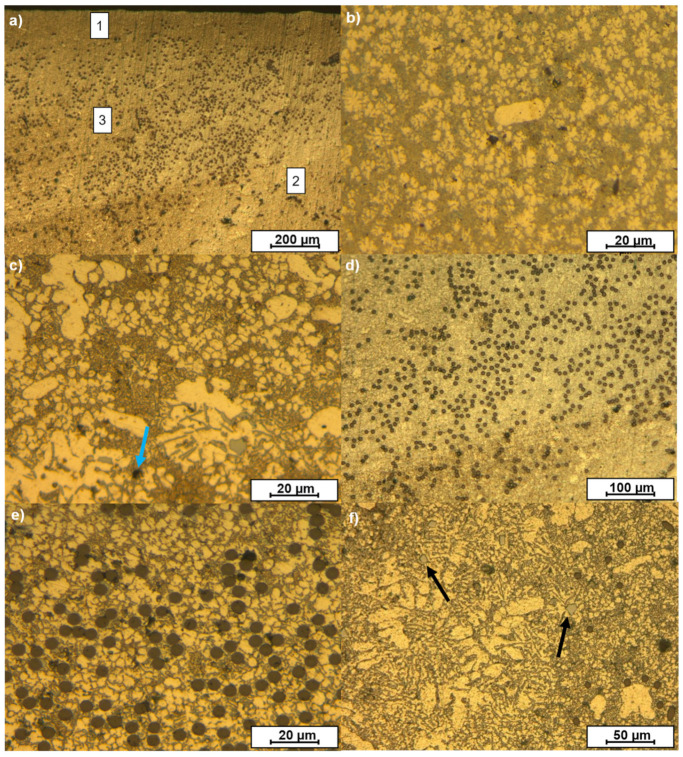
Microstructure of the sample obtained by optical microscopy: (**a**) general view of the cross-section of a sample reinforced with woven: zone 1 corresponds to the microstructure close to the die; zone 2 to the microstructure far from the die and the fibers and zone 3 to the microstructure close to the fiber; (**b**) zones in contact with the mold; (**c**) zones far from the mold and the fibers with arrowed pore; (**d**) samples with reinforcement in zones near the fibers; (**e**) magnification of (**d**,**f**) zones with arrowed α-AlFeMnSi phases.

**Figure 6 materials-15-03400-f006:**
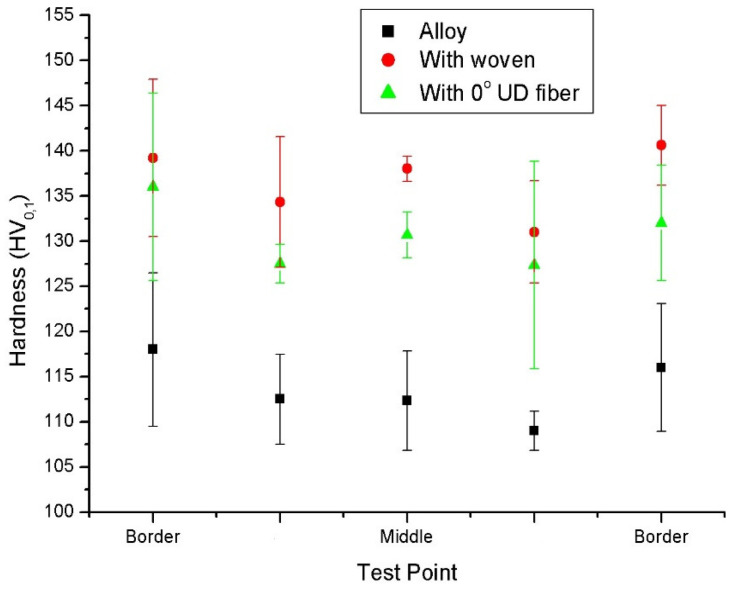
Vickers’s microhardness values measured on the cross-sections of the alloy and AMCs samples.

**Figure 7 materials-15-03400-f007:**
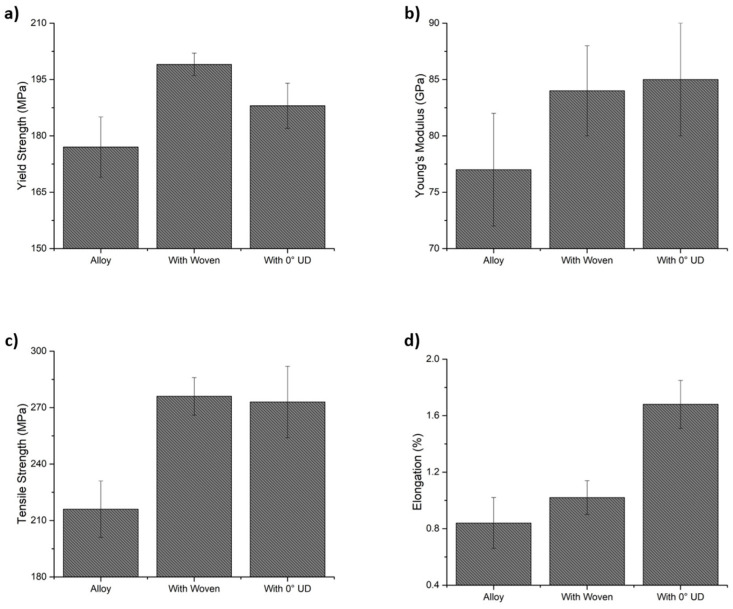
Results of the tensile tests: (**a**) yield strength; (**b**) Young’s modulus; (**c**) tensile strength; and (**d**) elongation at break.

**Figure 8 materials-15-03400-f008:**
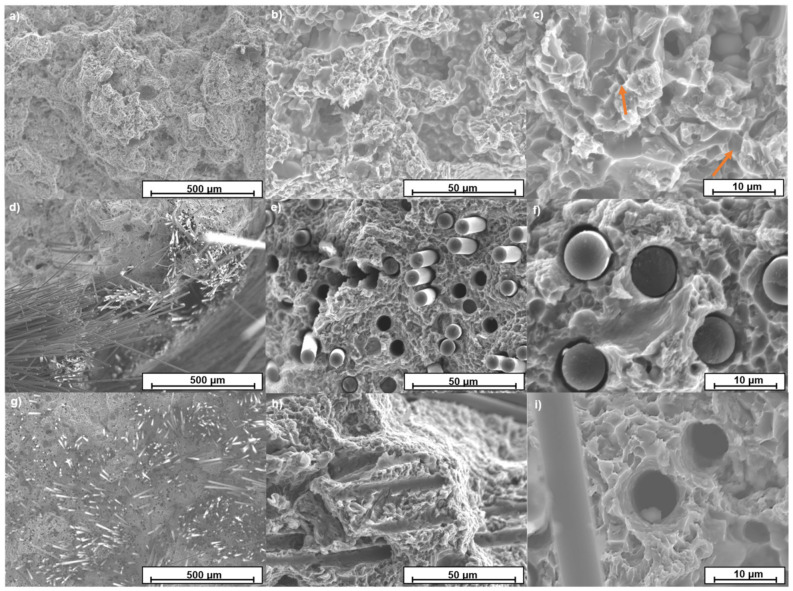
SEM fracture surfaces: (**a**) AA413 aluminum alloy and (**b**,**c**) magnification with dimples arrowed in orange; (**d**) aluminum reinforced with woven and (**e**,**f**) magnification of the extracted fibers; (**g**) aluminum reinforced with 0° unidirectional fiber and (**h**,**i**) magnification of different zones.

**Table 1 materials-15-03400-t001:** Samples manufactured.

Sample	Reinforcement	
Alloy	None	0%
Woven reinforced	0/90° 2 × 2 twill	8 vol.%
0° UD reinforced	0° UD	4 vol.%

**Table 2 materials-15-03400-t002:** Results of the mechanical properties.

Sample	Yield Strength (MPa)	Tensile Strength (MPa)	Elongation at Break (%)	Young’s Modulus (GPa)
Alloy	177 ± 8	216 ± 15	0.84 ± 0.18	77 ± 5
Woven reinforced	199 ± 3	276 ± 10	1.02 ± 0.12	84 ± 4
0° UD reinforced	188 ± 6	273 ± 19	1.68 ± 0.17	85 ± 5

**Table 3 materials-15-03400-t003:** Results of the mechanical models.

ROM (GPa)		HT (GPa)	
Al—Cf u	Al—Cf twill	Al—Cf u	Al—Cf twill
83.2	89.3	83.5	80.2

u: unidirectional reinforced composites; twill: bidirectional reinforced composite, woven.

## Data Availability

Not applicable.

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
