# Peer review of "Manufacturing of Aluminum Matrix Composites Reinforced with Carbon Fiber Fabrics by High Pressure Die Casting"

_materials, 2022, doi:10.3390/ma15093400_

Round 1
Reviewer 1 Report
The paper is very interesting and worth publication, however it requires some corrections. The questions and suggestions are listed below.- Was he microstructure of the material in Figs 3 and 5 were obtained by LM? Please provide appropriate comments in the text and in pictures captions.
- Too low magnification in Fig.5 does not allow for an unambiguous assessment of the information and for the differentiation of grain size in zones 1,2,3. With such low magnification, it is impossible to give such accurate values of the deviations from the mean diameter. The methodology for the evaluation of the stereological particle size has not been given. It is also difficult to interpret porosity from figures.
- In Fi. 7a the axis description is incorrect.
- The adopted models for the evaluation of the elastic properties of the composite should be verified. By applying the mixing rule (ROM), the values for the unidirectional reinforced composite are 83.2 GPa, and for the fabric reinforced composite 87.5 GPa. The assumptions of the Halapin-Tsai hypothesis proposed for the assessment should also be specified.
- Incorrect units for the value of modulus of elasticity expressed in MPa.
- Incorrect reference to Tab.4., which is not listed in the paper.
- The fracture microstructures presented in Fig.8 due to the low magnification and low quality of the pictures do not show the information described in the text. It is difficult to agree with the statement that the addition of a reinforcing fiber changes the type of fracture from brittle to ductile. It has not been specified for what type of composite was presented in Fig. 8g (it is probably a uni-directionally reinforced composite). For comparison, the fracture of fabric-reinforced composite should be included.
Author Response
Answer to reviewer comments.
1. Was the microstructure of the material in Figs 3 and 5 were obtained by LM? Please
provide appropriate comments in the text and in pictures captions.
These micrographs have been obtained by optical microscopy. We have added it to the text
in the different zones suggested.
2. Too low magnification in Fig.5 does not allow for an unambiguous assessment of the
information and for the differentiation of grain size in zones 1,2,3. With such low
magnification, it is impossible to give such accurate values of the deviations from the
mean diameter. The methodology for the evaluation of the stereological particle size has
not been given. It is also difficult to interpret porosity from figures.
We have added figures with higher magnifications and added the methodology to calculate
the grain size. There were no many pores, but we have arrowed in blue in figure 5c one of
them.
3. In Fig. 7a the axis description is incorrect.
We have changed the description.
4. The adopted models for the evaluation of the elastic properties of the composite should
be verified. By applying the mixing rule (ROM), the values for the unidirectional
reinforced composite are 83.2 GPa, and for the fabric reinforced composite 87.5 GPa. The
assumptions of the Halapin-Tsai hypothesis proposed for the assessment should also be
specified.
The reviewer is right. We have changed the ROM value and we have added some
considerations to the Halpin-Tsai model. We assume that we have an ideal solid with no
pores and that there is no interphase between matrix and reinoforcement. The form factor
was 30,000. The modulus used for the matrix was that measured for the unreinforced alloy.
5. Incorrect units for the value of modulus of elasticity expressed in MPa.
The reviewer is right. We have corrected the mistakes.
6. Incorrect reference to Tab. 4, which is not listed in the paper.
The reviewer is right. We have corrected to table 3.
7. The fracture microstructures presented in Fig. 8 due to the low magnification and low
quality of the pictures do not show the information described in the text. It is difficult to
agree with the statement that the addition of a reinforcing fiber changes the type of
fracture from brittle to ductile. It has not been specified for what type of composite was
presented in Fig. 8g (it is probably a uni-directionally reinforced composite). For
comparison, the fracture of fabric-reinforced composite should be included.
The reviewer is right. We have changed the figure with better imagesthat better show what
we observed. We have also included the images corresponding to the fabric composite.
The text has been modified to include an analysis on the new information provided.

Reviewer 2 Report
see attachment.

Author Response
Answer to reviewer comments.
1. Please succinctly clarify how carbon fiber improves mechanical properties in the Abstract.
We thank the reviewer for this point. We have added it in the abstract that there is an effective
load transfer between the fibers and the matrix.
2. The writing of the abstract section is well-structured. There are many studies on preparing
aluminum matrix composites by spark plasma sintering. The electromigration effect of pulsed
current during the spark plasma sintering process may benefit the wettability. Please add a
corresponding description in the appropriate place in the first paragraph of the introduction.
Here is some literature that can help with this process.
[1] Ding, Huaping, et al. "Enhancing strength-ductility synergy in an ex situ Zr-based metallic
glass composite via nanocrystal formation within high-entropy alloy particles." Materials &
design 210 (2021): 110108.
[2] Tan, Zhen, Wang, Lu, Xue, & Yunfei, et al. High-entropy alloy particle reinforced Al-based
amorphous alloy composite with ultrahigh strength prepared by spark plasma sintering.
Materials & Design, 109, 219-226. (10.1016/j.matdes.2016.07.086)
We thank the reviewer for this contribution. We have found interesting both papers and we
have added them in the introduction as well as an explanation of the mechanisms involved.
3. Will the increase of carbon fiber volume fraction affect the filling process of Al melt? Please
add some discussion about this.
Before this study, we made a parameter study with different volume fractions of fiber by adding
layers of carbon fiber. Although the porosity was not affected by the addition of only one layer,
we found that the addition of two or more layers of reinforcement increased dramatically the
number and size of pores. There were also, limitations in the infiltration of aluminum between
the layers. We have added this discussion to the document.
4. In the Conclusion section, the authors should give the best material composition they
prepared, along with the corresponding process parameters and performance data.
We thank the reviewer for this observation. We have added the suggested information in a new
conclusion in the conclusions section.
5. Are the mechanical properties of carbon fiber reinforced aluminum matrix composites
anisotropic? Please add some discussion about this.
Although in this work we have not made any study about the anisotropy of the fabricated
composites, there are several reasons to think that they have this property, particularly, the
aluminum reinforced with the unidirectional fibers. We have added this discussion to the text at
the end of section 4.2.

Reviewer 3 Report
Comments of authors
The paper entitled "Manufacturing of Aluminum Matrix Composites Reinforced with Carbon Fiber Fabrics by High Pressure Die Casting" is suitable for the materials Journal. The manuscript is exhaustively written and the results are supported by well prepared and discussed experimental data. The authors utilized carbon fiber in order to strengthen aluminum composites to improve its mechanical properties.
Some suggested papers for improving the wettability by coating and hot pressing
- Nour-Eldin, M., Elkady, O. & Yehia, H.M. Timeless Powder Hot Compaction of Nickel-Reinforced Al/(Al2O3-Graphene Nanosheet) Composite for Light Applications Using Hydrazine Reduction Method. J. of Materi Eng and Perform (2022). https://doi.org/10.1007/s11665-022-06749-w
- Elkady OA, Yehia H. M., Ibrahim A. A., Elhabak AM, Elsayed EM, Mahdy AA. Direct Observation of Induced Graphene and SiC Strengthening in Al–Ni Alloy via the Hot Pressing Technique. Crystals. 11(9):1142, 2021. https://doi.org/10.3390/cryst11091142

Author Response
Reviewer comments
Comments of authors
The paper entitled "Manufacturing of Aluminum Matrix Composites Reinforced
with Carbon Fiber Fabrics by High Pressure Die Casting" is suitable for the
materials Journal. The manuscript is exhaustively written and the results are supported
by well prepared and discussed experimental data. The authors utilized carbon fiber in
order to strengthen aluminum composites to improve its mechanical properties.
Some suggested papers for improving the wettability by coating and hot pressing
1. Nour-Eldin, M., Elkady, O. & Yehia, H.M. Timeless Powder Hot Compaction
of Nickel-Reinforced Al/(Al2O3-Graphene Nanosheet) Composite for Light
Applications Using Hydrazine Reduction Method. J. of Materi Eng and
Perform (2022). https://doi.org/10.1007/s11665-022-06749-w
2. Hossam M. Yehia· S. Allam, Hot Pressing of Al‑10 wt% Cu‑10 wt% Ni/x
(Al2O3–Ag) Nanocomposites at Different Heating Temperatures, metal and
materials international, 27, Pp. 500–513, (2021).
https://doi.org/10.1007/s12540-020-00824-4
3. Elkady OA, Yehia H. M., Ibrahim A. A., Elhabak AM, Elsayed EM, Mahdy
AA. Direct Observation of Induced Graphene and SiC Strengthening in Al–Ni
Alloy via the Hot Pressing Technique. Crystals. 11(9):1142, 2021.
https://doi.org/10.3390/cryst11091142
We thank the reviewer for the suggested papers. We found interesting all of them
and we have included them in the manuscript, which is richer now.

Reviewer 4 Report
It is a really interesting study about the matrix composites reinforced with carbon fiber, congratulations.
However the reviewer is not a native English speaking person, the language seems OK.
I have several questions, which needs to be answered mostly about the methods, I made those specific remarks, questions, comments in the manuscript_with_reviewers_comments
With proper corrections I think the manuscript will satisfy the publication criteria in Materials.

Author Response
Answer to reviewer comments.
The answer to the reviewer comments
Line 19. Up to 70 % of elongation.
We have revised the phrase which now is: “There was also a 70 % increase in elongation for
the unidirectionally reinforced samples, because of the finer microstructure and the load
transfer to the fibers, allowing the formation of larger voids in the matrix before breaking”
What about continuous infiltration of wire comp.? e.g.:
https://doi.org/10.4028/www.scientific.net/MSF.537-538.19
https://doi.org/10.4028/www.scientific.net/AST.50.1471
We thank the reviewer for the suggestions and we have incorporated them in the
introduction.
Line 130. If it had a dogbone shapes which dimensions are listed here? is it really necessary
to give it with 0.01 mm accuracy?
The reviewer is right, we have corrected the text.
Line 131. Size?
The sample had a dogbone shape. The size of the central zone was 55.0 mm x 10.0 mm x 2.0
mm. The total length of the samples was 116 mm. The carbon fibers were cut with a length
and width greater than the sample size.
We have corrected this in the text.
Line 132. How. Is it a horizontally opening die?
Yes, it is a horizontal opening die, we have included more details to explain the process.
Line 136. Was the die cold or preheated?
The mold was preheated to 280 °C. We have added this information to the text.
Line 157. Macro hardness tests would give that, or, carbon was dissolved in Al liquid
somehow?
We have observed that there was no dissolution of carbon in the liquid aluminum. However,
we have removed the details in this part to better organize the article.

Round 2
Reviewer 1 Report
Thank you for the correction. However, they are not entirely satisfactory. there is still a lack of appropriate magnifications on the structures and methods of image analysis with commonly used image analysis computer programs allowing to perform a stereological description of the grain size.
Reviewer 2 Report
Nice revision. I have no other questions.